# The Evaluation of Videos about Branched-Chain Amino Acids Supplements on YouTube ^TM^: A Multi-Approach Study

**DOI:** 10.3390/ijerph192416659

**Published:** 2022-12-11

**Authors:** Elif Günalan, Saadet Turhan, Betül Yıldırım Çavak, İrem Kaya Cebioğlu, Özge Çonak

**Affiliations:** 1Department of Nutrition and Dietetics, Istanbul Health and Technology University, Istanbul 34025, Turkey; 2Department of Occupational Therapy, Istanbul Health and Technology University, Istanbul 34025, Turkey; 3Institute of Graduate Education, Istinye University, Istanbul 34010, Turkey; 4Department of Nutrition and Dietetics, Yeditepe University, Istanbul 34755, Turkey; 5Department of Health Management, Beykent University, Istanbul 34398, Turkey

**Keywords:** dietary supplements, ergogenic aids, isoleucine, leucine, sports, valine

## Abstract

Branched-chain amino acids (BCAAs) are one of the most controversial ergogenic aids in terms of effectiveness and safety. This study aimed to evaluate the quality and reliability of BCAA supplements related to English videos on YouTube^™^ and to synthesize with the sentiment–emotion analysis of comments on videos. The content analysis of the information on videos was evaluated with the use of DISCERN, Journal of American Medical Association (JAMA) benchmark criteria, and Global Quality Score (GQS). In addition, word cloud and sentiment and emotional analysis of comments in videos were performed with the R package. As a result, the mean ± standard error values of DISCERN, JAMA, and GQS scores of all videos were 29.27 ± 1.97, 1.95 ± 0.12, and 2.13 ± 0.17, respectively. It was found that advertisement-free videos have a significantly higher DISCERN and GQS score than advertisement-included videos (*p* < 0.05). A moderately significant positive correlation was determined between DISCERN score of video content and the positive sentiment of video comments (rs: 0.400, *p* = 0.002). In conclusion, it was determined that BCAA-related YouTube^™^ videos have mostly very poor quality in terms of content and that videos with higher quality may receive positive comments from viewers according to the DISCERN instrument.

## 1. Introduction

### 1.1. Branched-Chain Amino Acids in Sports Nutrition

Branched-chain amino acids (BCAAs) are leucine, isoleucine, and valine amino acids metabolized primarily within the skeletal muscle [1]. BCAAs are found in different amounts in red meat and poultry (36.3%), milk and dairy products (16.7%), cheese (9.8%), bread, pasta, and cereals (17.3%), and legumes (4.8%) [2]. In addition, BCAA supplements are commercially available in different dosages (from 600 mg to 8000 mg per serving) and various ratios of leucine, isoleucine, and valine amino acids (usually in a 2:1:1 ratio) in sports markets [3]. Although the effects of BCAAs supplementation on various diseases, such as liver cirrhosis, renal failure, cancer, and also muscle wasting syndrome, have been investigated for more than 40 years in the literature, there is yet no consensus on its therapeutic effects [1].

Commercial BCAA supplements are commonly preferred in sports activities due to their muscle-related effects. BCAAs are claimed to have several functions as an ergogenic aid in sports, such as increasing muscle mass by activating protein synthesis, strengthening immunity, reducing soreness, improving cognitive function, and enhancing endurance capacity [4,5,6,7,8,9,10]. However, there are not enough studies in the literature to confirm most of these claims [4,5,6,7,8]. For instance, a double-blind controlled randomized study showed that 21 g of BCAA supplementation over a week does not enhance vertical jump performance in professional volleyball players [11]. According to another study, BCAA supplementation after a treadmill test did not provide any differences in the sensation of fatigue and plasma concentration of glucose, lactate, or ammonia [12]. In contrast, Gualano et al. (2011) demonstrated that BCAA supplementation increases exercise capacity and lipid oxidation [13]. The heterogeneity of the findings does not completely confirm the use of BCAAs as an ergogenic supplement for sports [14]. Currently, BCAAs are classified in the C group supplements, which do not have enough scientific evidence in terms of benefits for athletes, and no research has been conducted on them to provide a guideline by the Australian Institute of Sports (AIS) [3]. Consistently, recent guidelines do not mention that BCAA supplements are not seen as performance enhancers [15,16,17]. In addition, higher plasma concentrations of BCAAs can be related to risks of several chronic diseases, such as diabetes, cancer, and cardiovascular diseases (CVD) [18,19,20,21,22].

Although the effectiveness of BCAA supplementation as ergogenic aids remains unwarranted, most athletes tend to excessively consume BCAA supplements, which results in a multi-billion-dollar industry in sports supplement marketing [4]. Conflict of interest between scientific reality and financial concerns in this type of sports supplement could give rise to a potential threat to public health, especially in athletes. Determining the validity and reliability of motivation-related information regarding BCAA consumption is very important globally in terms of protecting the health of athletes.

### 1.2. Literature Review Regarding Social Media Analysis

Globalization and digitalization enable the development of new strategies for marketing sports supplements and expanding the market volume [23]. In this context, social media is one of the most popular marketing strategies for supplements among athletes [24,25,26]. Social media platforms are extremely low-cost, instructive, and easily accessible tools that provide a variety of nutrition-related information for athletes [27]. Social media can shape consumer attitudes with easier access to knowledge in sports nutrition [28]. In this way, it also has a significant role in promoting the usage of sports supplements. For instance, the study by Wiens et al. demonstrated that one of the educational tools for dietary supplement usage for young Canadian athletes is the internet [29]. Another study claimed that social media contributes to more effective communication between sports nutritionists and athletes with mobile and visual learning [30].

Especially with the pandemic period, the use of social media platforms has increased, and the accuracy of the information shared on these platforms has gained importance. Several tools have been developed to determine the content quality of web-based information, such as DISCERN, Journal of American Medical Association (JAMA) benchmark criteria, and Global Quality Score (GQS). To this point, some studies are notable regarding the evaluation of the quality and reliability of shared knowledge in the various websites [31,32,33,34]. For instance, Ng et al. investigated the quality of online dietary and herbal supplement websites (*n* = 87), according to the DISCERN instrument, and they found high variability in the DISCERN instrument scores across websites [31]. In another study, the information about *Ephedra sinica* on online websites (*n* = 28) had a poor quality [33]. Similarly, Wahab et al. demonstrated, on the basis of a modified DISCERN tool, that websites (*n* = 321) containing information about *Eurycoma longifolia*, which is an herbal medicine, were generally low-quality [34]. These studies show that there may be lower quality and reliability in online sources about dietary supplements with commercial value. This case indicates that there may be potential misinformation in online resources for sports supplements promoted by commercial concerns.

This strategy is also used to determine the contents of the videos shared on the YouTube^™^ platform, the world’s largest and most well-known video-sharing platform, to inform various patient groups in terms of quality, usefulness, and reliability [35,36,37,38,39]. Although, until today, billions of videos regarding sports nutrition and the use of sports supplements have been shared on YouTube^™^, there has been no study to determine the content analysis of these videos. This situation expresses the potential gap in providing reliable, valid, and high-quality education tools to the public. This may cause unconscious use of supplements, such as BCAAs, for which there is no consensus on their use in sports, marketed by sources that have no data concerning their scientific value and accuracy for athletes. Because of this, the scientific value and reliability of the BCAA-related video content should be evaluated via eligible instruments to properly guide video viewers.

On the other hand, the major concern in studies evaluating the quality and reliability of content information on the internet is that users’ responses to these contents are not determined [23]. YouTube^™^ is not only a video-sharing tool but also a platform that allows its users to comment on videos. With a text mining method, sentiment and emotion analysis of the comments shared by the users of videos can also be performed, and this problem can be solved easily. It enables the analysis of the comments in the high data volume on social media platforms to determine the opinions of the user population on that subject [40]. Although this approach has been widely used in political and social sciences, it has also begun being used in health and sports-related fields in recent years [41]. For instance, Eyipınar et al. investigated the sentiment analysis of 1902 comments on Turkish YouTube^™^ videos regarding sports nutrition. According to that investigation, it was determined that 27.6% of the comments obtained from YouTube^™^ videos about sports nutrition were positive, 17.3% were negative, and the remainder were neutral [42]. In another study, sentiment analysis was performed on Twitter comments regarding the most-often discussed sports supplements, such as creatine, BCAA, whey protein, nitric oxide, and multivitamins. As a result of the study, positive sentiments have been determined to be more intense in all selected supplements [23]. On the other hand, emotional information, such as anger, fear, anticipation, trust, surprise, sadness, joy, and disgust, regarding video comments can be obtained by the NRC Emotion Lexicon [43]. However, there has been no study on the emotional analysis of sports nutrition or supplements.

### 1.3. Aim of the Study

When the current literature was evaluated in the scope of YouTube^™^ English videos regarding BCAA supplements, it was determined that neither a reliable and valid study of video content nor a sentiment and emotional analysis of BCAA-related video comments has been conducted. Considering the potential health risks related to online misinformation, the current study aimed to (1) evaluate the quality and reliability of the information on BCAAs-related videos, (2) determine the sentiment and emotion analysis of comments of viewers as a response to videos, (3) synthesize data from these qualitative and quantitative methods, and (4) understand viewers responses to different ranged quality videos.

The use of YouTube^™^ as an effective tool in sharing information on sports nutrition marketing has made the evaluation of the qualitative and quantitative data regarding shared content inevitable. It is considered that the application of this approach, especially for supplements such as BCAAs, whose efficacy and safety are not fully known, will play an important role in protecting the health of athletes through the conscious use of supplements and professionalizing of digitalized educational tools. It will also provide a scientific perspective on the sports supplement industry without ties to financial concerns.

## 2. Materials and Methods

### 2.1. Study Design

The current study focuses on the videos about BCAA supplements on the YouTube^™^ social media platform. “BCAA” and “BCAA supplements” terms were searched on YouTube^™^ as keywords, and the most-viewed 100 videos were obtained for each term. Out-of-topic, non-English, duplicated, non-verbal, and poor pronunciation videos were excluded from the study. At the end of the process, 55 videos were recruited for qualitative and quantitative analysis. A flow chart of the video selection process is presented in Figure 1.

The duration of the videos, the number of likes and views, the collected replies, and the job of the video uploader were retrieved. The quality and reliability assessments of video contents were completed using the DISCERN instrument, JAMA benchmark criteria, and GQS in collaboration with a registered dietitian (B.Y.Ç) and physiotherapist (S.T). In addition, the word cloud and sentiments–emotions analysis of comments were performed with the R package.

### 2.2. Assessment Tools of Video Reliability, Validity, and Quality


*Discern Instrument*


The DISCERN tool is an assessment scale developed for patients and providers to assess the reliability and quality of information [44,45,46]. The tool, which consists of 16 items in total, is divided into 3 parts. Items 1 through 8 form the first part and measure the reliability of the information. Items 9 through 15 form the second part, measuring the quality of the information, and the last section consists of a single item with an overall quality rating (item 16). DISCERN uses a 5-point Likert scale. For evaluating the first 15 items, 1 point means “no”, and 5 points means “yes”; and the responses are evaluated within this range. For the 16th item, 1 point means “low quality with serious or extensive deficiencies”, and 5 points means “high quality with minimum-wax deficiencies”, and the responses are evaluated within this range [47,48]. Detailed information is available in Appendix A [49]. The total DISCERN score was calculated as the sum of the first 15 items and can be a minimum of 15 and a maximum of 75. The reliability and quality of the information are characterized by an increase in scores, where a score of 17–27 points denotes “very poor”, 28–38 points denotes “poor”, 39–50 points denotes “medium”, 51–62 points denotes “good”, and 63–75 points denotes “excellent” [50]. DISCERN is freely accessible at http://www.discern.org.uk/ (accessed on 15 September 2022) [51].


*JAMA Benchmark Criteria*


JAMA benchmark criteria instrument is one of the leading tools used to evaluate the medical information obtained from online sources [38]. It includes 4 criteria—authorship, attribution, disclosure, and currency—of 1 point, each with a total score of 4 points. Detailed information about JAMA criteria is presented in Appendix A. In the JAMA evaluation, 0–1 point represents insufficient information, 2–3 points represents partially sufficient information, and 4 points represents completely sufficient information [35,36].


*Global Quality Score*


GQS is a scoring system defined by Bernard et al., which can be used to assess a video in terms of its instructive aspects for viewers [37,52]. It allows us to evaluate the quality, streaming, and ease of use of the information presented in online videos (Appendix A). In the evaluation of GQS, a score of one point indicates that the video has the poorest quality and is not at all useful for viewers, while a score of five points indicates that the video has excellent quality and is very useful for viewers [52,53].

### 2.3. Sentiment–Emotion Analysis of Videos with R Programming Language

Sentiment analysis of 55 YouTube^™^ videos included in BCAA-related research was conducted using the statistical program R 4.1.3, an open-source software frequently used in text mining research, with packages designed to predict the strength of positive and negative sentiments [54]. R provides rich libraries containing many packages in many applications, such as statistical calculations, machine learning, and deep learning. Sentiment analysis was carried out using the Natural Language Process (NLP), one of the machine learning techniques, on the dataset created by the noisy nature of the video comments collected with the authorization of the YouTube^™^ API. After obtaining authorization to use YouTube^™^ API with the R programming, the comments of the related videos were turned into a corpus with package functions. Within the scope of the research, in collecting data for the analysis of video comments and making the data suitable for sentiment analysis, “devtools”, “vosonSML”, “magrittr”, and “tuber” packages are used, which combine many common tasks, facilitate analysis, and collect data from social media networks [55,56,57,58]. In the sentiment analysis of the dataset, “tm”, “syuzhet”, “tidytext”, and “tidyverse” packages were utilized [59,60,61,62]. “dplyr”, “ggplot2”, “SnowballC”, “wordcloud”, and “RColorBrewer” packages were used to visualize the analyses [63,64,65,66,67]. Within the scope of the research, a total of 25 767 comments from the videos constitute the dataset. The dataset comprised video comments and relationships, such as sayings and replies between users. In the analysis conducted with the R programming language, the words “can”, “will”, “be”, “sir”, “just”, “take”, and “video” were excluded from the dataset because they were not in the context of the research. After appropriate preprocessing and feature extraction, the dataset was classified using the NRC Emotion Lexicon. According to the NRC Emotion Lexicon, eight basic emotions (anger, fear, anticipation, trust, surprise, sadness, joy, and disgust) and two sentiments (negative and positive) comprise a list of words and their associations [54]. With this classification, the sentiment and emotion scores of the comments in YouTube^™^ videos were detected and converted to percentage distribution. Detailed information about the workflow of sentiment–emotion analysis is presented in Figure 2.

### 2.4. Statistical Analysis

SPSS software version 22 was used for statistical analysis (SPSS Inc., Chicago, IL, USA). The normality distribution of variables was carried out using the Shapiro–Wilk test. Since none of the metric variables followed a normal distribution, they were presented with the median and interquartile range (IQR), while categorical variables were displayed as percentages (%). The means of quantitative data with non-normal distribution were compared with the Kruskal–Wallis and Mann–Whitney U Test. The relationship between variables was examined by calculating Spearman’s correlation coefficient. A *p*-value < 0.05 was considered as statistically significant.

## 3. Results

### 3.1. General Characteristics of the BCAA Videos

The general properties of evaluated videos are presented in Table 1. According to the table, the median (IQR) values of the number of views, likes, comments, and collected replies on videos were 121,925.00 (70,594–325,302), 1700.00 (494–3700), 192.00 (73–356), and 57.00 (16–143), respectively. The percentages of advertisement-included and advertisement-free videos were 63.6% and 36.4%, respectively. Expertise designations of the video uploader were fitness trainer (49.1%), nutritionist (1.8%), bodybuilder (12.7%), amateur athlete (5.5%), fitness influencer (7.3%), manufacturer (1.8%), athlete/life/nutrition coach (1.8%), physician (3.6%), and anonymous (16.4%).

### 3.2. Validity and Reliability Analysis of the BCAA Videos by DISCERN, JAMA, and GQS Scoring Systems

Descriptive statistics of scoring systems are shown in Table 2. The mean ± standard error values of DISCERN, JAMA, and GQS scores of all videos were 29.27 ± 1.97, 1.95 ± 0.12, and 2.13 ± 0.17, respectively. According to DISCERN scores, 58.2% (n = 32) of videos were very poor quality, 21.8% (n = 12) of videos were poor quality, 14.5% (n = 8) of videos were average quality, and 5.5% (n = 3) of videos were good quality. On the other hand, the JAMA benchmark criteria analysis demonstrated that, of the 55 videos, 36.4% (20 videos) achieved only 1 criterion and had insufficient information in the video contents, 56.4% (31 videos) achieved 2–3 criteria, and had partially sufficient information in video content, and 7.3% (4 videos) achieved all 4 criteria, and had completely sufficient information in video content. Lastly, of the 55 videos assessed, 65.5% (n = 36) were of low quality, 16.4% (n = 9) were of intermediate quality, and 18.2% (n = 10) were of high quality, according to the GQS scores. In addition, advertisement-free videos had significantly higher DISCERN (*p* = 0.007) and GQS (*p* = 0.026) scores when compared with advertisement-included videos (Table 2).

When a comparison of DISCERN, JAMA, and GQS scores was conducted on the basis of who uploaded the videos, significant differences were observed in scores in terms of expertise designations of the video uploader (*p* = 0.20, *p* = 0.24, *p* = 0.22, respectively). In this context, the dietitian (n = 1) had the highest scores of DISCERN, JAMA, and GQS as video uploaders (45.0, 4.0, 4.0, respectively). Physician (n = 2) uploaded the second-highest-scoring videos (34.0 ± 2.8, 3.5 ± 0.7, 3.0), while those uploaded by the manufacturer (n = 1) scored the lowest (17.0, 2.0, 1.0), according to DISCERN, JAMA, and GQS scores, respectively.

### 3.3. Analysis of Video Comments

The number of times a word was used in the comments and the word cloud constructed from them is presented in Figure 3. According to those results, BCAA (n = 3854), protein (n = 3378), BCAAs (n = 2297), workout (n = 1846), good (n = 1824), like (n = 1650), whey (n = 1609), thanks (n = 1508), muscle (n = 1422), and need (n = 1355) were determined as most-often used words in total video comments (n = 25,767).

As presented in Figure 4, the percentage distributions of sentiments in the video comments were determined as 70% positive and 30% negative, while emotions were distributed as trust (25.1%), joy (15.6%), anticipation (15.4%), sadness (10.4%), fear (9.0%), surprise (8.7%), disgust (8.3%), and anger (7.4%).

### 3.4. Synthesis of Features of Videos and Sentiment Scores with DISCERN, JAMA, and GQS Scores

Table 3 presented correlation findings between features of videos and sentiment–emotion scores with DISCERN, JAMA, and GQS scores. According to those results, there were significant moderate correlations between the total number of likes, comments, and collected replies with DISCERN (rs: 0.446, *p* = 0.001; rs: 0.476, *p* < 0.001; rs: 0.507, *p* < 0.001, respectively) and GQS scores (rs: 0.472, *p* < 0.001; rs: 0.468, *p* < 0.001; rs: 0.563, *p* < 0.001, respectively) of videos. Lastly, a moderately significant positive correlation was determined between DISCERN score of video content and the positive sentiment of video comments (rs: 0.400, *p* = 0.002).

## 4. Discussion

The present study showed that BCAA-related YouTube^™^ videos have very mostly poor quality in terms of content, and higher quality content may be related to positive comments from viewers, according to the DISCERN instrument. To the best of our knowledge, this study has unique value, being different from other studies in the literature in terms of including two different approaches and synthesizing mixed methods. From this aspect, it is an innovative study of the current literature on sports nutrition and the health of athletes.

The majority of the videos included in this study have been uploaded to YouTube^™^ by fitness trainers (49.1%) and bodybuilders (12.7%). In the literature, the main resource of information regarding the type, use, and utility of sports supplements has been shown to be diverse (sports trainers, nutritionists, coaches, etc.) depending on the sports branches, sex, and competition level [24,25,26]. For instance, the main source of information for amateur and professional rugby players to consume dietary supplements was a sports trainer (25.8%) [25]. In another study carried out among Turkish footballers, nutritionists (29.7%) were demonstrated to be the main motivator for dietary supplement consumption [24].

Regarding the quality and reliability of the information, videos were mostly of poor quality, according to DISCERN and GQS instruments, while they had partially sufficient information, according to the JAMA benchmark criteria. This situation could be related to video uploaders and the inclusion or absence of advertisements. Regarding video uploaders, the percentage of videos uploaded by nutritionists (1.8%), who are experts in sports supplements and nutrition, was quite low. As the major source of nutritional knowledge, a nutritionist provides a better grasp of dietary patterns and the use of dietary supplements with a high degree of scientific value [68,69]. Therefore, the fact that there are few nutritionists among the video uploaders can be associated with the low scientific quality of the videos. In the current study, the DISCERN, JAMA, and GQS scores of nutritionists and physicians were higher than that of sports supplement manufacturers as video uploaders. In addition, advertisement-included videos had significantly lower DISCERN and GQS scores. Consistently, several studies have reported that the information available on commercial websites for dietary and herbal supplements is largely poor quality compared with health portal websites [31,70]. Similar to these studies, Gkouskou et al. declared that commercial websites had a significantly lower reliability score than institutional and other websites [71]. This may be related to the fact that commercial concerns can contribute to the lack of scientific accuracy and validity.

In the literature, there were not enough studies analyzing the quality and reliability analysis of online information about sports supplements. This situation limits the comparison of DISCERN, JAMA, and GQS scores with similar studies. However, there are a few studies that analyzed the DISCERN scores of online resources regarding herbal supplements, sports medicine, and nutrition [31,70,71,72,73]. For instance, Ng et al. showed that online resources containing complementary and alternative-medicine-specific information vary in quality, according to the DISCERN instrument [72]. Similarly, quality assessment studies of websites regarding dietary and herbal supplements showed that high variabilities are observed in DISCERN scores. The average total DISCERN scores with standard deviations in these studies were determined as 44.80 ± 11.53 and 47.64 ± 10.38, respectively [31,70]. The present study showed that the mean ± standard error value of DISCERN score was 29.27 ± 1.97. In another study, the quality of medical information on four different topics, “Mediterranean diet”, “sports nutrition”, “nutrition, dysphagia, and children”, and “herbs and common cold” was evaluated from the web pages. According to their results, the category of “sports nutrition” had the lowest score from the first part of the DISCERN instrument. Nevertheless, there were no statistically significant differences between the categories [71]. The fact that our findings are similar to those in the literature confirms our data.

Concerning the sentiment analysis of the video comments, a strongly positive attitude of YouTube^™^ users was noted regarding BCAA supplement videos. In addition, good (n = 1824), like (n = 1650), and great (n = 1211), which are the most-used words in video comments, support this finding. The investigated comments could be related to the potential effects of BCAAs, such as activating protein synthesis, reducing soreness, enhancing cognitive activity, boosting immunity, and improving endurance capacity [4,5,6,7,8,9,10]. In fact, the words protein (n = 3378) and muscle (n = 1422) are often emphasized in video comments. However, no research or consensus has been carried out to provide a guideline regarding conscious consuming of BCAAs [4,5,6,7,8]. Therefore, this situation may be related to scientific gaps in social media platforms on sports nutrition.

Surprisingly, the results of similar studies carried out on sentiment analysis of social media comments about BCAA are contradictory. For instance, Catalani et al. conducted a sentiment analysis of 18,595 comments about sports supplements. In the study, it was noted that the comments about supplements were generally positive. Similar to our outcomes, it has been reported that 87% of users have positive feelings about BCAAs [23]. On the other hand, Eyipınar et al. performed sentiment analysis on the comments of six different YouTube^™^ videos on sports nutrition using a text mining technique. As a result of their study, it was determined that YouTube^™^ users have predominantly negative thoughts about the consumption of BCAA supplements [42]. The reasons for the differences between the studies may depend on the social media content, users, time, the number of videos, and uploaders of the videos.

In another study, 2708 dietary supplements on Amazon.com were grouped and identified as potentially unsafe supplements from users’ comments by topic modeling techniques, specifically a variation of latent Dirichlet allocation (LDA) [74].

According to our findings, the most frequently observed emotions in BCAA-related videos were trust (25.1%), joy (15.6%), and anticipation (15.4%). It can be considered an unexpected result for BCAAs because BCAA supplements are accepted as one of the most discussed supplements in terms of safe use, benefits, and efficacy. There is not enough scientific evidence for the use of BCAAs in sports-related activity. Moreover, the effects of long-term unconscious consumption of BCAA supplements remain mostly unknown, and high serum concentrations of BCAAs may be associated with risks of various chronic diseases [18,19,20,21,22]. In this context, Canela et al. have demonstrated that higher concentrations of baseline BCAAs are related to the increased risk of CVD [18]. Similarly, Tobias et al. claimed that BCAAs are associated with the burden of cardiometabolic diseases in the population [19]. Lastly, it has been reported that increased plasma BCAA levels are related to a more than twofold elevated risk of developing pancreatic cancer later in life [20].

In the present study, an interesting outcome concerns the correlations between the total number of likes, comments, and collected replies with DISCERN and GQS scores of videos, which were not observed in the JAMA benchmark criteria. Consistently, a moderate correlation was found between positive sentiment scores and only DISCERN scores. This situation supports that clear, accurate, and fluent videos lead to more likes, positive emotions, and more comments by users. On the other hand, the reason JAMA scores of videos did not correlate with any parameters may be related to the fact that JAMA focuses more on the scientific value of the knowledge via determining authorship, attribution, disclosure, and currency. The scientific value of videos may not be understood by the public/YouTube^™^ users.

With the transition to the digital age and globalization, social networks and online resources are starting to be used more and more as a primary source of information in determining the efficacy and safety of sports supplements [23]. Moreover, the pandemic has contributed to this process, especially with the quarantine period [75]. However, information that may affect public health may be presented on the Internet by people who are not experts in the field. The poor quality of BCAA supplement videos on YouTube^™^ in our research supports this situation. Considering the potential health risks regarding online misinformation for most-discussed supplements, validated and reliable knowledge about sports nutrition should be presented by health professionals, particularly nutritionists [71]. Thus, online platforms can be considered useful training tools for athletes in the topics of sports nutrition in the future.

Reliable use of online information resources as educational material is possible by determining the quality of the content. There are several important issues to be questioned on this point: (1) How accurate, reliable, and usable is the information? (2) How does this information affect the users? (3) What is the interaction between the information and the user? The determination of reliability and validity of online knowledge does not provide information about the opinions and attitudes of users toward this information. This deficiency has been discussed in the literature before [23], but it has been eliminated with the sentiment and emotion analysis approach in the current study. Sentiment analysis provides the determination of opinions of individuals/consumers regarding a product, service, or process. This process was carried out by text mining methods that are generally used intensively in social sciences [41], while it has not been studied sufficiently in the fields of sports, athlete health, and nutrition [42]. However, considering that social networks are being used more widely among athletes, it is clear that much more work can be conducted in the future. As a starting point, this study design can be an innovative model for future research.

### Strengths and Limitations of the Study

This study has several strengths when compared with other studies in the literature. First, the DISCERN, JAMA, and GQS scores of videos on sports supplements have not been evaluated before in studies in the literature. Secondly, comments on sports supplements were evaluated in terms of only positive, negative, and neutral attitudes in previous studies [23,42]. However, the current study presents a wide range of data about eight basic emotions (trust, joy, anticipation, sadness, fear, surprise, disgust, and anger) and two sentiments (positive and negative). Thirdly, our study has unique value compared with the other studies in the literature in terms of combining scoring of DISCERN, JAMA, and GQS with sentiment–emotion analysis.

On the other hand, the current study has some limitations that must be considered when evaluating the outcomes. Initially, the investigated videos were selected among the most viewed during the research period. Therefore, the effects of temporal alterations may restrict the repeatability of the study for the same videos. In addition, the study is limited to only YouTube^™^. Different information, posts, and comments about BCAA supplements can be found on other platforms such as Facebook, Twitter, Instagram, etc. This situation makes it difficult to evaluate the study through all social media networks. Lastly, this study includes only English videos regarding BCAA supplements, which could be considered another limitation. In the future, content and comment analysis can be planned concerning online resources in different languages for all sports supplements.

## 5. Conclusions

### 5.1. Theoretical Implications

This study demonstrated that YouTube^™^ videos concerning BCAAs were generally of very poor quality and had partially sufficient information. In particular, advertisement-included videos had lower scientific accuracy and quality. In addition, commentators of the videos had mostly positive sentiments and trust emotions. There were significantly moderate correlations between the total number of likes, comments, and collected replies with DISCERN and GQS scores of videos. Nevertheless, it has been shown that a higher DISCERN score can give rise to more positive comments from video viewers. Considering the potential health risks related to online misinformation for most-discussed sports supplements, such as BCAA, for athletes, the reliability, validity, and quality of knowledge in online resources should be investigated more.

### 5.2. Practical Implications

This study presents an innovative model for future social media research. According to the study, both content and user comments should be taken into account in sports nutrition studies. Thanks to the multi-approach design, the quality of the shared information and the attitude of the social media user can be easily monitored. Uploading information about sports nutrition to social media should be provided or checked by health professionals. Thus, online resources may be more reliable and useful for athletes and other individuals. In fact, social media platforms, especially YouTube^™^, can be accepted as a part of professional education and research in the developing digitalized world.

This view is based on the fact that sports nutritionists share, within their social contexts, how much they manage their bodies and how they perceive potential health risks when using BCAAs. This is important for users, clinicians, and professionals, as well as manufacturers who need new regulations regarding their production and marketing processes. It will ensure the planned growth of this area, which is open to development in the many fields of the supplements industry.

## Figures and Tables

**Figure 1 ijerph-19-16659-f001:**
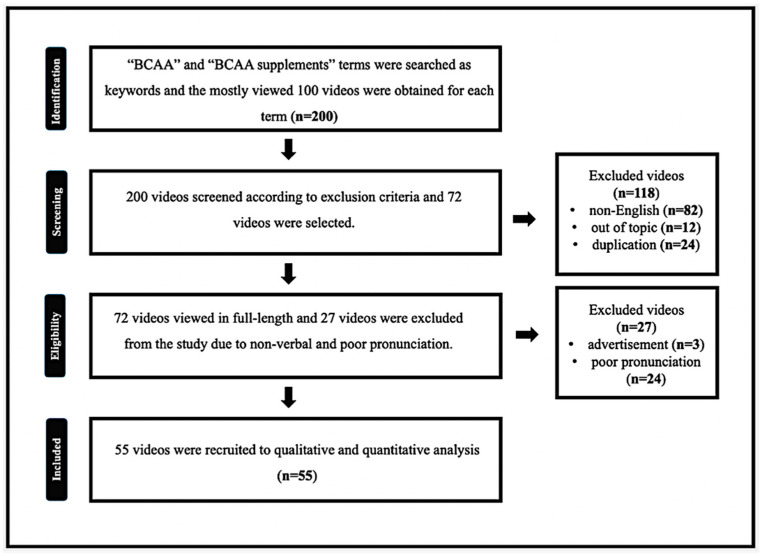
Flow chart of video selection procedure according to eligibility criteria.

**Figure 2 ijerph-19-16659-f002:**
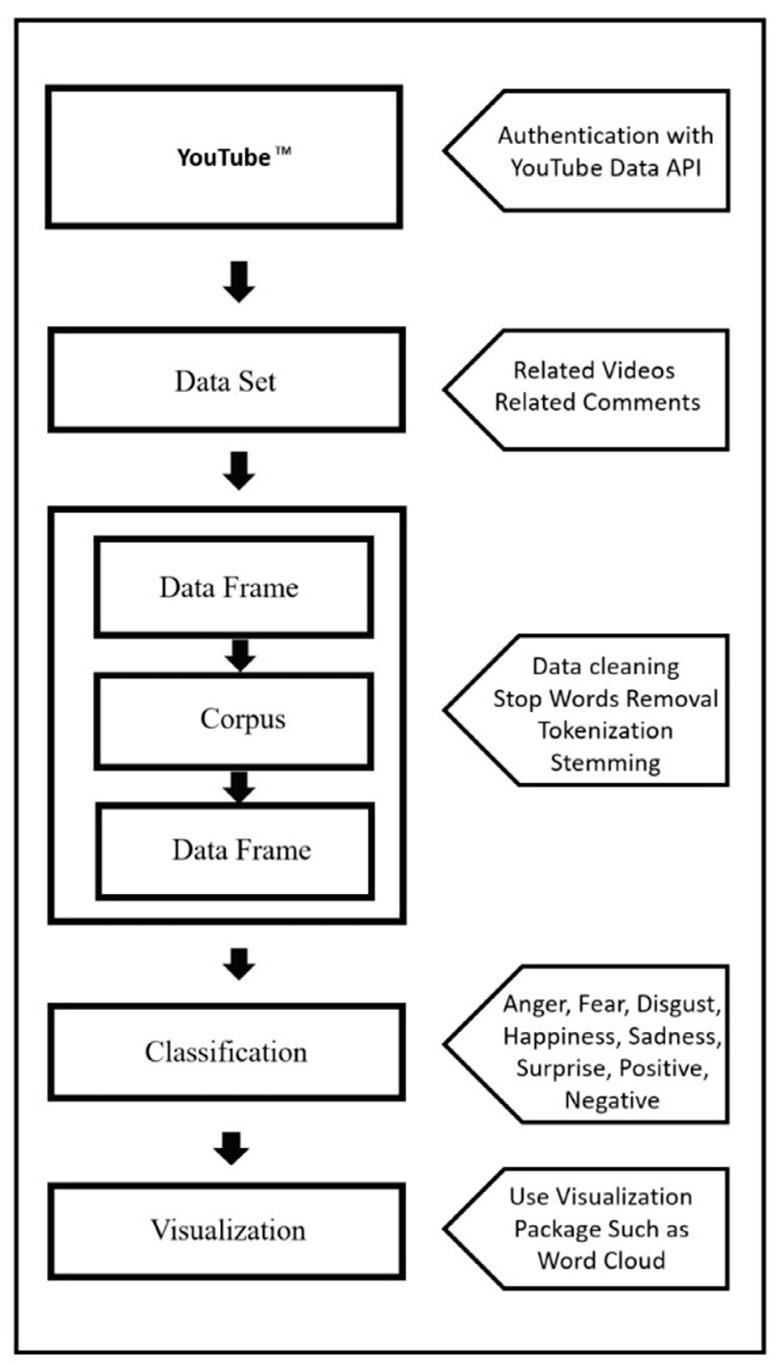
Workflow of sentiment–emotion analysis of the video comments.

**Figure 3 ijerph-19-16659-f003:**
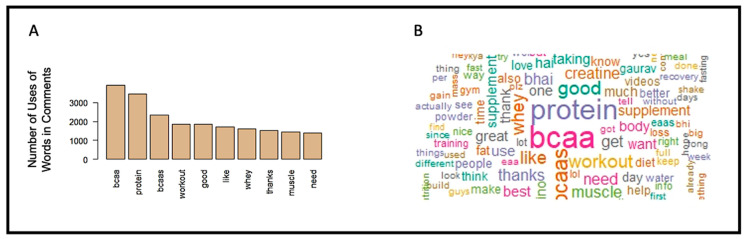
(**A**) Number of uses of words in comments. (**B**) Word cloud of most-often used words in comments.

**Figure 4 ijerph-19-16659-f004:**
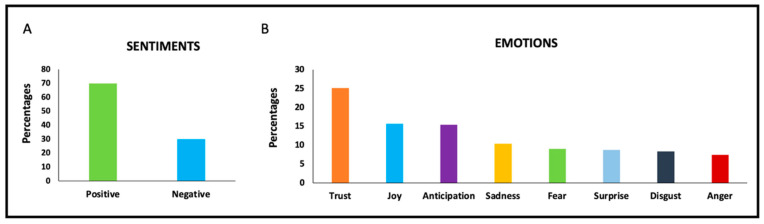
Percentages of sentiments (**A**) and emotions (**B**) of comments in videos.

**Table 1 ijerph-19-16659-t001:** General characteristics of selected videos.

	Median (IQR)	Mean ± S.E.
Video Length (seconds)	353.00 (249–536)	416.58 ± 34.94
Total Number of Views	121,925.00 (70,594–325,302)	386,334.02 ± 130,873.74
Total Number of Likes	1700.00 (494–3700)	7432.95 ± 3224.10
Total Number of Comments	192.00 (73–356)	472.40 ± 140.77
Collected Replies	57.00 (16–143)	108.45 ± 24.97

IQR: interquartile range.

**Table 2 ijerph-19-16659-t002:** Descriptive statistics of scoring systems regarding advertisements.

	Overall Videos (*n* = 55)		Ads Included (*n* = 35)		No Ads Included (*n* = 20)		Sig.
	Median (IQR)	Mean ± S.E.	Median (IQR)	Mean ± S.E.	Median (IQR)	Mean ± S.E.	*p*
DISCERN	27.0 (22–36)	29.27 ± 1.37	24.00 (19–31)	26.71 ± 1.61	30.50 (25.3–41.8)	33.75 ± 2.20	0.007 *
JAMA	2.00 (1–2)	1.95 ± 0.12	2.00 (1–3)	2.00 ± 0.17	2.00 (1–2)	1.85 ± 0.17	0.780
GQS	2.00 (1–3)	2.13 ± 0.17	1.00 (1–3)	1.86 ± 0.19	2.50 (1–4)	2.6 ± 0.28	0.026 *

IQR: interquartile range, JAMA, Journal of American Medical Association; GQS: Global Quality Score; * *p* < 0.05.

**Table 3 ijerph-19-16659-t003:** Correlations between features of videos and sentiment–emotion scores with the DISCERN, JAMA, and GQS scoring systems.

		DISCERN		JAMA		GQS	
		r_s_	*p*	r_s_	*p*	r_s_	*p*
Features of Videos	Video Length (seconds)	0.214	0.117	−0.114	0.408	0.150	0.273
	Total Number of Views	0.346	0.010 *	−0.053	0.702	0.379	0.004 *
	Total Number of Likes	0.446	0.001 *	0.061	0.657	0.472	0.000 *
	Total Number of Comments	0.476	0.000 *	−0.094	0.496	0.468	0.000 *
	Collected Replies	0.507	0.000 *	0.008	0.951	0.563	0.000 *
Sentiments & Emotions	Positive	0.400	0.002 *	0.189	0.167	0.384	0.004 *
	Negative	0.363	0.006 *	0.125	0.363	0.281	0.038 *
	Trust	0.394	0.003 *	0.170	0.215	0.341	0.011 *
	Joy	0.263	0.052	0.207	0.130	0.255	0.060
	Anticipation	0.256	0.059	0.255	0.060	0.200	0.143
	Sadness	0.289	0.033 *	0.122	0.374	0.259	0.057
	Fear	0.207	0.129	0.053	0.701	0.145	0.292
	Surprise	0.264	0.051	0.247	0.070	0.215	0.114
	Disgust	0.245	0.072	−0.041	0.766	0.149	0.278
	Anger	0.088	0.521	0.050	0.716	0.064	0.643

r_s_: 0.40–0.59 indicates a moderate positive correlation; * *p* < 0.05.

## Data Availability

Data described in the manuscript can be available upon request pending application and approval.

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
