# Peer review of "The Evaluation of Videos about Branched-Chain Amino Acids Supplements on YouTube™: A Multi-Approach Study"

_ijerph, 2022, doi:10.3390/ijerph192416659_

Round 1
Reviewer 1 Report
The main aim of the paper was to examine the branched-chain amino acids (BCAAs) that is a popular ergogenic aids among athletes due to their role in stimulating protein synthesis, recovery processes from exercise, and protection of mental health after prolonged exercise. However, the study proposes that unconscious intake of BCAA supplements may contribute to the progression of pathological conditions such as renal failure and several types of cancer. Hence the aim of the study aimed to evaluate the quality and reliability of BCAA supplements related to English videos on YouTube and to conduct sentiment analysis of comments on videos. The study revealed that advertisement-free videos have a significantly higher DISCERN and GQS score than advertisement-included videos. Additionally, there were also correlations between the total number of likes, comments and collected replies with DISCERN. Hence, these findings make an original contribution since the study revealed Hence, these findings make an original contribution since the study revealed that BCAA-related YouTube videos generally have poor content quality, whereas better quality videos receive more positive comments from viewers.
The theoretical foundation on which the research was based was lacking and should be introduced in the Introduction and then significantly expanded in the literature review. The authors should take as a starting-point one or more sufficiently contrasted theories and apply them to this new context of analysis to justify the need to develop this new research. The paper should incorporate a more solid argumentation that allows justification of the reason for the selection of the explanatory variables that are considered in the empirical analysis. Stating that there is a lack of theoretical explanations and empirical evidence is an insufficient reason/justification to conduct the research. Furthermore, as far as possible, the theoretical framework should be sufficiently solid to justify that the relevant variables that should explain the phenomenon under study are those considered in the analysis and only those variables. In short, why is this research necessary; why are you using the theoretical model; what are the research gaps; and what recent justification have you provided for the aforementioned? In summary, fully explain the theoretical framework (in the Introduction and Literature review sections) that served as the foundation for your conceptual model that you developed.
Additionally, the Introduction section was much too short and did not cite a sufficient number of sources. The Introduction typically provides the background to the research problem and lays the groundwork by citing the origin of the idea/problem. There are generally several sections that provide a natural flow:
Overview / background information: Provide an overview / background information on all of the different variables of your research title in this section! In what context (industry area) does the problem find itself and the importance of the sector under investigation?
Area of concern / topic importance: Clarify the area of concern, or whatever needs to justify the research. Indicate why you believe that it is, in fact, a researchable problem, i.e. is the problem researchable and is it feasible? Do the preliminary data and available resources support its feasibility? Why is the topic important to discipline, country, region and globe?
Research gaps (as mentioned above): Have others studied this area - is there gap worth exploring? What are the research gaps, identified by others (in-text reference all sources, since this provides justification for your research? Use multiple in-text referencing/citations to collectively highlight a number of studies that identified potential gaps in terms of the topic.
Mandate more research is needed: Find journal articles/theses that mandate more research is needed in your area of study/topic by reading the “Introduction” and “Limitations and future research” sections, which will show possible research gaps / new research areas. Example of possible research gaps (all sources should be in-text referenced): Additional research should be conducted in different countries; different samples / target groups / unit of analysis; different industries / institutions; different research approaches (qualitative vs. quantitative); larger samples sizes / or more representative samples; new theoretical model, etc.! This section should be expanded upon throughout the literature review.
Lastly, end the “Introduction/background to the research problem” section with a statement that provides the main aim of your study that would serve to address the abovementioned research gaps.
Theoretical Framework (Literature Review) section was also lacking. The theoretical framework helps the researcher to explain time tested theories that embody the findings of numerous investigations on how phenomena occur under given scenarios. The theoretical framework can be written in prose (written format) and in a visual format (diagram) reinforced by a summarized explanation of the diagram. The theoretical framework frames the study and identifies the key theories and concepts that underpin your study. Stated differently, the theoretical framework provides background information on your study, which must emanate from a theoretical model in the field of study. Your discourse should explain the “how” and “why” things have transpired in the manner that they have, to anchor the research and thesis. Your narrative must include examples of the theoretical underpinning, where it has been used and by who, as well as a summary of their findings. Hence, your theoretical framework needs to be made more apparent, and your review of literature section should also be significantly expanded.
The was NO reference list? However, ensure that you provide a number of recent academic sources (e.g. journal articles) from 2020 – 2022, which should be incorporated throughout the paper to support the research, since there is much recent research on YouTube.
The Material and Methods section was apt.
The Results and Discussion sections were appropriately presented. However, I could not check the age of your sources since you omitted the reference list, so please be sure to supplement the Discussion section with recent literature sources as mentioned above if this is not the case.
There Conclusions section was very brief, so you should also include a separate Theoretical Implication section and a separate Managerial or Practical Implications’ section as subheadings in the Conclusion section as follows:
- Theoretical implications: What contribution has the research has made to theory based on the theoretical foundation/model that you selected (or in your case did not select).
- Managerial or practical implications: Explain the contribution that the research has made in terms of managerial or practical implications.
Overall, an interesting paper, which makes an original contribution, but I suggest some major revisions before the paper can be accepted for publication, and it is very important that you include your reference list!
Author Response
Reviewer 1:
The main aim of the paper was to examine the branched-chain amino acids (BCAAs) that is a popular ergogenic aids among athletes due to their role in stimulating protein synthesis, recovery processes from exercise, and protection of mental health after prolonged exercise. However, the study proposes that unconscious intake of BCAA supplements may contribute to the progression of pathological conditions such as renal failure and several types of cancer. Hence the aim of the study aimed to evaluate the quality and reliability of BCAA supplements related to English videos on YouTube and to conduct sentiment analysis of comments on videos. The study revealed that advertisement-free videos have a significantly higher DISCERN and GQS score than advertisement-included videos. Additionally, there were also correlations between the total number of likes, comments and collected replies with DISCERN. Hence, these findings make an original contribution since the study revealed Hence, these findings make an original contribution since the study revealed that BCAA-related YouTube videos generally have poor content quality, whereas better quality videos receive more positive comments from viewers.
-Thanks for your valuable attention and opinions.
The theoretical foundation on which the research was based was lacking and should be introduced in the Introduction and then significantly expanded in the literature review. The authors should take as a starting-point one or more sufficiently contrasted theories and apply them to this new context of analysis to justify the need to develop this new research. The paper should incorporate a more solid argumentation that allows justification of the reason for the selection of the explanatory variables that are considered in the empirical analysis. Stating that there is a lack of theoretical explanations and empirical evidence is an insufficient reason/justification to conduct the research. Furthermore, as far as possible, the theoretical framework should be sufficiently solid to justify that the relevant variables that should explain the phenomenon under study are those considered in the analysis and only those variables. In short, why is this research necessary; why are you using the theoretical model; what are the research gaps; and what recent justification have you provided for the aforementioned? In summary, fully explain the theoretical framework (in the Introduction and Literature review sections) that served as the foundation for your conceptual model that you developed.
-Thanks for your valuable contributions, introduction and literature review was updated.
Additionally, the Introduction section was much too short and did not cite a sufficient number of sources. The Introduction typically provides the background to the research problem and lays the groundwork by citing the origin of the idea/problem. There are generally several sections that provide a natural flow:
Overview / background information: Provide an overview / background information on all of the different variables of your research title in this section! In what context (industry area) does the problem find itself and the importance of the sector under investigation?
-Thanks for your valuable contributions. The problem is given as clearly in the text.
Area of concern / topic importance: Clarify the area of concern, or whatever needs to justify the research. Indicate why you believe that it is, in fact, a researchable problem, i.e. is the problem researchable and is it feasible? Do the preliminary data and available resources support its feasibility? Why is the topic important to discipline, country, region and globe?
-Thanks for your comments. The area of interest/subject importance has been expanded.
Research gaps (as mentioned above): Have others studied this area - is there gap worth exploring? What are the research gaps, identified by others (in-text reference all sources, since this provides justification for your research? Use multiple in-text referencing/citations to collectively highlight a number of studies that identified potential gaps in terms of the topic.
-Thanks for your contributions. This section is revised according to your guideline.
Mandate more research is needed: Find journal articles/theses that mandate more research is needed in your area of study/topic by reading the “Introduction” and “Limitations and future research” sections, which will show possible research gaps / new research areas. Example of possible research gaps (all sources should be in-text referenced): Additional research should be conducted in different countries; different samples / target groups / unit of analysis; different industries / institutions; different research approaches (qualitative vs. quantitative); larger samples sizes / or more representative samples; new theoretical model, etc.! This section should be expanded upon throughout the literature review.
-Thanks for valuable contributions. New references have been added to these sections. However, neither a reliable and valid study of video content nor a sentiment and emotional analysis of BCAA-related video comments was conducted in the literature. Therefore, we were only able to draw more attention to gaps and potential hazards in the literature.
Lastly, end the “Introduction/background to the research problem” section with a statement that provides the main aim of your study that would serve to address the abovementioned research gaps.
-Thanks for your comments. According to your comments, this section was updated.
Theoretical Framework (Literature Review) section was also lacking. The theoretical framework helps the researcher to explain time tested theories that embody the findings of numerous investigations on how phenomena occur under given scenarios. The theoretical framework can be written in prose (written format) and in a visual format (diagram) reinforced by a summarized explanation of the diagram. The theoretical framework frames the study and identifies the key theories and concepts that underpin your study. Stated differently, the theoretical framework provides background information on your study, which must emanate from a theoretical model in the field of study. Your discourse should explain the “how” and “why” things have transpired in the manner that they have, to anchor the research and thesis. Your narrative must include examples of the theoretical underpinning, where it has been used and by who, as well as a summary of their findings. Hence, your theoretical framework needs to be made more apparent, and your review of literature section should also be significantly expanded.
- Theoretical Framework is enlarged and updated.
According to your all guideline, Introduction section is revised. According to that:
1.1. Branched-Chain Amino Acids In Sports Nutrition
Branched-chain amino acids (BCAAs) are leucine, isoleucine, and valine amino acids, metabolized primarily within the skeletal muscle [1]. BCAAs are found in different amounts in red meat and poultry (36.3%), milk and dairy products (16.7%), cheese (9.8%), bread, pasta and cereals (17.3%), and legumes (4.8%) [2]. Apart from that, BCAA supp-lements are commercially available in different dosages (from 600 mg to 8000 mg per serving) and various ratios of leucine, isoleucine, and valine amino acids (usually in a 2:1:1 ratio) in sports markets [3]. Although the effects of BCAAs supplementation on various diseases such as liver cirrhosis, renal failure, cancer, and also muscle wasting syndrome have been investigated for more than 40 years in the literature, there is no consensus on its therapeutic effects yet [1].
Commercial BCAA supplements are commonly preferred in sports activities due to their muscle-related effects. BCAAs are claimed to have several functions as an ergogenic aid in sports, such as increasing muscle mass by activating protein synthesis, strengthening immunity, reducing soreness, improving cognitive function, and enhancing endurance capacity [4-10]. However, there are not enough studies in the literature to confirm most of them [4-8]. For instance, a double-blind controlled randomized study has shown that 21 g of BCAA supplementation over a week does not enhance vertical jump performance in professional volleyball players [11]. According to another study, BCAA supplementation after a treadmill test has not provided any differences in the sensation of fatigue and plasma concentration of glucose, lactate, and ammonia [12]. In contrast, Gualano et al. (2011) have demonstrated that BCAA supplementation increases exercise capacity and lipid oxidation [13]. The heterogeneity of the findings does not completely confirm the use of BCAA as an ergogenic supplement for sports [14]. Already, BCAAs are classified in the C group supplements which do not have enough scientific evidence in terms of benefits for athletes or no research carried out to provide a guideline, by the Australian Institute of Sports (AIS) [3]. Consistently, recent guidelines do not mention the BCAA supplements or BCAA supplements are not seen as performance enhancers [15-17]. In addition, higher plasma concentrations of BCAAs can be related to risks of several chronic diseases such as diabetes, cancer and cardiovascular diseases (CVD) [18-22].
Although the effectiveness of BCAA supplementation as ergogenic aid remains unwarranted, most athletes tend to excessive consumption the BCAA supplement and it refers to a multi-billion dollar industry in sports supplement marketing [4]. Conflict of interest between scientific reality and financial concerns in this type of sports supplement could give rise to a potential threat to public health, especially in athletes. Determining the validity and reliability of BCAA consumption motivation related information is very important as globally in terms of protecting the health of athletes.
1.2. Literature Review Regarding Social Media Analysis
Globalization and digitalization enable the development of new strategies for the marketing of sports supplements and expanding the market volume [23]. In this context, social media is one of the most popular marketing strategies for supplements among athletes [24-26]. Social media platforms are an extremely low-cost, instructive, and easily accessible tool that provides a variety of nutrition-related information for athletes [27]. Social media can shape consumer attitudes with easier access to knowledge in sports nutrition [28]. In this way, it also has a significant role in promoting the usage of sports supplements. For instance, the study of Wiens et al. has demonstrated that one of the educational tools of dietary supplement usage is determined as the internet in young Canadian athletes [29]. Another study has claimed that social media contribute to more effective communication between sports nutritionist and athlete with mobile and visual learning [30].
Especially with the pandemic period, the use of social media platforms has increased and the accuracy of the information shared on these platforms has gained importance. Several tools have been developed to determine the content quality of web-based information such as DISCERN, Journal of American Medical Association (JAMA) benchmark criteria, and Global Quality Score (GQS). In this point, some studies are notable regarding the evaluation of the quality and reliability of shared knowledge in the various websites [31-34]. For instance, Ng et al. have investigated the quality of online dietary and herbal supplement websites (n=87) according to DISCERN instrument and they found high variability in DISCERN instrument scores across websites [31]. In another study, the information about Ephedra sinica on online websites (n=28) had a poor-quality [33]. Similarly, Wahab et al. have demonstrated that websites (n=321) containing information about Eurycoma longifolia which is a herbal medicine, generally were low-quality based on a modified DISCERN tool [34]. These studies show that there may be lower quality and reliability in online sources about dietary supplements with commercial value. This case indicates that there may be potential misinformation in online resources for sports supplements promoted by commercial concerns.
This strategy is also used to determine the contents of the videos shared on the Youtube platform the world's largest and most well-known video-sharing platform, to inform various patient groups, in terms of quality, usefulness, and reliability [35-39]. Although until today billions of videos regarding sports nutrition and the use of sports supplements have been shared on YouTube, there was no study to determine the content analysis of these videos. This situation expresses the potential gap in providing reliable, valid, and high-quality education tools to the public. This may cause to unconscious use of supplements, such as BCAAs, for which there is no consensus on their use in sports, by sources that have no data about scientific value and accuracy for athletes. According to that, the scientific value and reliability of the BCAA-related video content should be emerged via eligible instruments, to properly guide video viewers.
On the other hand, the major concern in studies evaluating the quality and reliability of content information on the internet is that users' responses to these contents are not determined [23]. YouTube is not only a video-sharing tool but also a platform that allows its users to comment on videos. With a text mining method, sentiment and emotion analysis of the comments shared by the users of videos can also be performed and this problem can be solved easily. It enables the analysis of the comments in the high data volume on social media platforms to determine the opinions of the user population on that subject [40]. Although this approach has been widely used in political and social sciences, it has also been started practiced in health and sports-related fields in recent years [41]. For instance, Eyipınar et al. investigated the sentiment analysis of 1902 comments on Turkish YouTube videos regarding sports nutrition. According to that, it was determined that 27.6% of the comments obtained from YouTube videos about sports nutrition were positive, 17.3% were negative and the rest were neutral [42]. In another study, sentiment analysis has been performed on Twitter comments about the most discussed sports supplements such as creatine, BCAA, whey protein, nitric oxide, and multivitamin. As a result of the study, positive sentiments have been determined as more intense in all of the selected supplements [23]. On the other hand, emotional information such as anger, fear, anticipation, trust, surprise, sadness, joy, and disgust, regarding video comments can be obtained by the NRC Emotion Lexicon [43]. However, there was no study on the emotional analysis of sports nutrition or supplements.
1.3. Aim of the Study
When the current literature was evaluated in the scope of YouTube English videos about BCAA supplements, it was determined that neither a reliable and valid study of video content nor a sentiment and emotional analysis of BCAA-related video comments was conducted. Considering the potential health risks related to online misinformation, the current study aimed to evaluate the quality and reliability of the information on BCAA-related videos (1), to determine the sentiment and emotion analysis of comments of viewers as a response to videos (2), to synthesize data from these qualitative and quantitative methods (3), and to understand viewers responses to different ranged quality videos (4).
The use of YouTube as an effective tool in sharing information on sports nutrition marketing has made it inevitable to evaluate the qualitative and quantitative data regarding shared content. It is considered that the application of this approach, especially for supplements such as BCAA, whose efficacy and safety are not fully known, will play an important role in protecting the health of athletes through the conscious use of supplements and professionalizing digitalized educational tools. It will also provide a scientific perspective on the sports supplement industry without financial concerns.
The was NO reference list? However, ensure that you provide a number of recent academic sources (e.g. journal articles) from 2020 – 2022, which should be incorporated throughout the paper to support the research, since there is much recent research on YouTube.
- The reference list of the article was in word format. It has now been added to both pdf and word forms in the revised manuscript. In addition, the majority of references are in the date range you specify.
The Material and Methods section was apt.
-Thanks for your valuable contributions.
The Results and Discussion sections were appropriately presented. However, I could not check the age of your sources since you omitted the reference list, so please be sure to supplement the Discussion section with recent literature sources as mentioned above if this is not the case.
-Thanks for your comments. The age of my sources is checked and recent studies have been used.
There Conclusions section was very brief, so you should also include a separate Theoretical Implication section and a separate Managerial or Practical Implications’ section as subheadings in the Conclusion section as follows:
- Theoretical implications: What contribution has the research has made to theory based on the theoretical foundation/model that you selected (or in your case did not select).
- According to your comments, related alterations were performed. Thanks for your contributions. Revised Section:
Theoritical Implications:
This study demonstrated that YouTube videos about BCAAs were generally of very poor quality and had partially sufficient information. Especially, advertisement-included videos had lower scientific accuracy and quality. In addition, commentators of the videos had mostly positive sentiments and trust emotions. There were significantly moderate correlations between the total number of likes, comments, and collected replies with DISCERN and GQS scores of videos. Nevertheless, it has been shown a higher DISCERN score can give rise to more positive comments from video viewers. Considering the po-tential health risks related to online misformation for mostly discussed sports supplements such as BCAA, etc. for athletes, the reliability, validity, and quality of knowledge in online resources should be more investigated.
- Managerial or practical implications: Explain the contribution that the research has made in terms of managerial or practical implications.
- According to your comments, related alterations were performed. Thanks for your contributions. Revised Section:
Practical Implications:
This study presents an innovative model for future social media research. According to that, both content and user comments should be taken into account in sports nutrition studies. Thanks to multi-approach design, the quality of the shared information and the attitude of the social media user can be easily monitored. Uploading information about sports nutrition to social media should be provided or checked by health professionals. Thus, online resources may be more reliable and useful for athletes and other individuals. By the way, social media platforms especially YouTube can be accepted as a part of the professional education and research in the developing digitalized world.
It is based on the fact that sports nutritionists share with their social contexts how much they manage their bodies and how they perceive potential health risks when using BCAAs. This is important for users, clinicians, and professionals, as well as manufacturers who need new regulations regarding their production and marketing processes. It will ensure the planned growth of this area, which is open to development in the many fields of the industry.
Overall, an interesting paper, which makes an original contribution, but I suggest some major revisions before the paper can be accepted for publication, and it is very important that you include your reference list!
-Thanks for your valuable contributions. According to your comments, the manuscript and reference list have been revised.

Reviewer 2 Report
Dear authors,
after reading the paper proposed by you, I consider that the following improvements are necessary for the paper to be considered for publishing:
- the abstract is too long, please present only the general conclusions, the information that is relevant for potential readers
- "Internet is a one of the most popular marketing strategies" - channel, maybe some ideas on social media marketing would bring more light on the subject
- Eyipınar et al. - the resource is unmarked - line 79 + conclusions
- "This instrument is a 5-point Likert-type scale consisting of an overall quality assessment (item 16) plus 15 items" - needs more clarity
- this section "Sentiment-Emotion Analysis of Videos with R Programming Language" must be described in a more friendly manner for a better understanding of the packages used
- While categorical variables were displayed as percentages (%). - line 164 - the sentence is unclear
- for the section Analysis of Video Comments - I consider that it would be interesting to select the Top 10 most relevant words, and exclude verbs like can, will.... - for me is not clear how these words can give a relevant and intelligible image of the selected comments. Or maybe to analyse these words from a perspective related to their relevance for the subject of the research - these are the most used words, but are they the most relevant?
- a comparison of the content of videos depending on who uploaded the video would be interesting
Success!
Author Response
Point 1: after reading the paper proposed by you, I consider that the following improvements are necessary for the paper to be considered for publishing:
Response 1: Thanks for your valuable contributions.
Point 2: the abstract is too long, please present only the general conclusions, the information that is relevant for potential readers
Response 2: Thanks for your contributions. The abstract was shortened, and unnecessary information for potential readers was excluded.
Point 3: "Internet is a one of the most popular marketing strategies" - channel, maybe some ideas on social media marketing would bring more light on the subject
Response 3: Thanks for your valuable contributions. Based on your comment, this section has been updated and more referenced..
Point 4: Eyipınar et al. - the resource is unmarked - line 79 + conclusions
Response 4: Thanks for your contributions. Resources were updated according to your guideline.
Point 5: "This instrument is a 5-point Likert-type scale consisting of an overall quality assessment (item 16) plus 15 items" - needs more clarity
Response 5: Thanks for your contributions. DISCERN instrument was revised in the text. According to that:
The DISCERN tool is an assessment scale developed for patients and providers to assess the reliability and quality of information [44-46]. The tool, which consists of 16 items in total, is divided into 3 parts. Items 1 and 8 form the first part and measure the reliability of the information. While forming the second part between the 9th and 15th items and measuring the quality of the information; the last section consists of a single item with an overall quality rating (item 16). DISCERN is a 5-point Likert scale. While evaluating the first 15 items, 1 point means “no,” and 5 points means “yes” and are evaluated within this range. In the 16th item, 1 point means “low quality with serious or extensive deficiencies” and 5 points “high quality with minimum-wax deficiencies” and are evaluated within this range [47, 48]. Detailed information is available in Table S1 [49]. The total DISCERN score is calculated as the sum of the first 15 items and can be a minimum of 15 and a maximum of 75. The reliability and quality of the information are characterized by an increase in scores, between 17-27 points "very poor," 28-38 points "poor," 39-50 points "medium," 51-62 points "good," between 63-75 points " excellent” [50]. DISCERN is freely accessible at http://www.discern.org.uk/ [51].
Point 6: this section "Sentiment-Emotion Analysis of Videos with R Programming Language" must be described in a more friendly manner for a better understanding of the packages used
Response 6: Thanks for your contributions. Related section was revised in the manuscript according to your guidance.
Point 7: While categorical variables were displayed as percentages (%). - line 164 - the sentence is unclear
Response 7: Thanks for your contributions. This sentence was revised to “Since none of the metric variables showed normal distribution, they were presented with the median and interquartile range (IQR), while categorical variables were displayed as percentages (%)….”
Point 8: for the section Analysis of Video Comments - I consider that it would be interesting to select the Top 10 most relevant words, and exclude verbs like can, will.... - for me is not clear how these words can give a relevant and intelligible image of the selected comments. Or maybe to analyse these words from a perspective related to their relevance for the subject of the research - these are the most used words, but are they the most relevant?
Response 8: Thanks for your valuable comments. The related sections were revised in the text and figure.
Point 9: a comparison of the content of videos depending on who uploaded the video would be interesting
Response 9: Distribution of expertise branches of video uploaders was extremely heterogeneous, and some of them were not enough in terms of statistical analysis. Therefore, we only carried out the Kruskal Wallis test but not the Mann-Whitney U test. According to that, it was observed significant differences in scores between expertise branches of the video uploader (p=0.20, p=0.24, p=0.22, respectively). In this context, the dietitian (n = 1) has the highest scores of DISCERN, JAMA, and GQS as video uploaders (45.0, 4.0, 4.0, respectively). Physician (n = 2) uploaded the second-highest-scoring videos (34.0 ± 2.8, 3.5 ± 0.7, 3.0), while those uploaded by the manufacturer (n = 1) scored the lowest (17.0, 2.0, 1.0) in terms of DISCERN, JAMA, and GQS scores, respectively.

Round 2
Reviewer 1 Report
The main aim of the paper was to examine the branched-chain amino acids (BCAAs) that is a popular ergogenic aids among athletes due to their role in stimulating protein synthesis, recovery processes from exercise, and protection of mental health after prolonged exercise. However, the study proposes that unconscious intake of BCAA supplements may contribute to the progression of pathological conditions such as renal failure and several types of cancer. Hence the aim of the study aimed to evaluate the quality and reliability of BCAA supplements related to English videos on YouTube and to conduct sentiment analysis of comments on videos. The study revealed that advertisement-free videos have a significantly higher DISCERN and GQS score than advertisement-included videos. Additionally, there were also correlations between the total number of likes, comments and collected replies with DISCERN. Hence, these findings make an original contribution since the study revealed. Hence, these findings make an original contribution since the study revealed that BCAA-related YouTube videos generally have poor content quality, whereas better quality videos receive more positive comments from viewers.
-No response or action was required from the authors for this comment.
The theoretical foundation on which the research was based was lacking and should be introduced in the Introduction and then significantly expanded in the literature review. The authors should take as a starting-point one or more sufficiently contrasted theories and apply them to this new context of analysis to justify the need to develop this new research. The paper should incorporate a more solid argumentation that allows justification of the reason for the selection of the explanatory variables that are considered in the empirical analysis. Stating that there is a lack of theoretical explanations and empirical evidence is an insufficient reason/justification to conduct the research. Furthermore, as far as possible, the theoretical framework should be sufficiently solid to justify that the relevant variables that should explain the phenomenon under study are those considered in the analysis and only those variables. In short, why is this research necessary; why are you using the theoretical model; what are the research gaps; and what recent justification have you provided for the aforementioned? In summary, fully explain the theoretical framework (in the Introduction and Literature review sections) that served as the foundation for your conceptual model that you developed.
-The contributions, introduction and literature review was suitably updated as requested in response to this comment.
Additionally, the Introduction section was much too short and did not cite a sufficient number of sources. The Introduction typically provides the background to the research problem and lays the groundwork by citing the origin of the idea/problem. There are generally several sections that provide a natural flow. Overview / background information: Provide an overview / background information on all of the different variables of your research title in this section! In what context (industry area) does the problem find itself and the importance of the sector under investigation?
-The Introduction section was aptly expanded and context was five to the overview / background information on all of the different variables were provided/highlighted.
Area of concern / topic importance: Clarify the area of concern, or whatever needs to justify the research. Indicate why you believe that it is, in fact, a researchable problem, i.e. is the problem researchable and is it feasible? Do the preliminary data and available resources support its feasibility? Why is the topic important to discipline, country, region and globe?
-The area of interest/subject importance was expanded as recommended.
Research gaps (as mentioned above): Have others studied this area - is there gap worth exploring? What are the research gaps, identified by others (in-text reference all sources, since this provides justification for your research? Use multiple in-text referencing/citations to collectively highlight a number of studies that identified potential gaps in terms of the topic.
-The research gaps were also expanded as suggested.
Mandate more research is needed: Find journal articles/theses that mandate more research is needed in your area of study/topic by reading the “Introduction” and “Limitations and future research” sections, which will show possible research gaps / new research areas. Example of possible research gaps (all sources should be in-text referenced): Additional research should be conducted in different countries; different samples / target groups / unit of analysis; different industries / institutions; different research approaches (qualitative vs. quantitative); larger samples sizes / or more representative samples; new theoretical model, etc.! This section should be expanded upon throughout the literature review.
-New references were included and the research gaps were highlighted as well as potential risks in the literature according to my comment.
Lastly, end the “Introduction/background to the research problem” section with a statement that provides the main aim of your study that would serve to address the abovementioned research gaps.
-The research problem was appropriately updated as per my suggestions.
Theoretical Framework (Literature Review) section was also lacking. The theoretical framework helps the researcher to explain time tested theories that embody the findings of numerous investigations on how phenomena occur under given scenarios. The theoretical framework can be written in prose (written format) and in a visual format (diagram) reinforced by a summarized explanation of the diagram. The theoretical framework frames the study and identifies the key theories and concepts that underpin your study. Stated differently, the theoretical framework provides background information on your study, which must emanate from a theoretical model in the field of study. Your discourse should explain the “how” and “why” things have transpired in the manner that they have, to anchor the research and thesis. Your narrative must include examples of the theoretical underpinning, where it has been used and by who, as well as a summary of their findings. Hence, your theoretical framework needs to be made more apparent, and your review of literature section should also be significantly expanded.
-The theoretical framework literature was aptly updated and expanded based on my suggestions.
There was NO reference list? However, ensure that you provide a number of recent academic sources (e.g. journal articles) from 2020 – 2022, which should be incorporated throughout the paper to support the research, since there is much recent research on YouTube.
- The reference list was included as requested and the sources were sufficiently current.
The Material and Methods section was apt.
-No response or action was required from the authors for this comment.
The Results and Discussion sections were appropriately presented. However, I could not check the age of your sources since you omitted the reference list, so please be sure to supplement the Discussion section with recent literature sources as mentioned above if this is not the case.
-The age of my sources was sufficiently recent in this section.
There Conclusions section was very brief, so you should also include a separate Theoretical Implication section and a separate Managerial or Practical Implications’ section as subheadings in the Conclusion section as follows:
Theoretical implications: What contribution has the research has made to theory based on the theoretical foundation/model that you selected (or in your case did not select).
According to your comments, related alterations were performed. Thanks for your contributions. Revised Section:
Managerial or practical implications: Explain the contribution that the research has made in terms of managerial or practical implications.
-The Theoretical implications and Practical implications sections were expanded as suggested
I detected one spelling error, so it is suggested that you read through the paper again, as well as run it through a “Spelling and Grammar” check to establish if there other errors.
Page 13 line 461: “5.1. Theoritical Implications” should be “5.1. Theoretical Implications”
Overall, an interesting paper, which makes an original contribution, and I am generally satisfied with the author changes, so the paper can now be accepted for publication. However, it is important the one last a “Spelling and Grammar” check is conducted before the paper is published.
Author Response
Dear Reviewer; thanks for your valuable contributions and comments. According to your comments, required corrections was performed. Detailed information was presented at the below.
Reviewer 1: The main aim of the paper was to examine the branched-chain amino acids (BCAAs) that is a popular ergogenic aids among athletes due to their role in stimulating protein synthesis, recovery processes from exercise, and protection of mental health after prolonged exercise. However, the study proposes that unconscious intake of BCAA supplements may contribute to the progression of pathological conditions such as renal failure and several types of cancer. Hence the aim of the study aimed to evaluate the quality and reliability of BCAA supplements related to English videos on YouTube and to conduct sentiment analysis of comments on videos. The study revealed that advertisement-free videos have a significantly higher DISCERN and GQS score than advertisement-included videos. Additionally, there were also correlations between the total number of likes, comments and collected replies with DISCERN. Hence, these findings make an original contribution since the study revealed Hence, these findings make an original contribution since the study revealed that BCAA-related YouTube videos generally have poor content quality, whereas better quality videos receive more positive comments from viewers.
No response or action was required from the authors for this comment.
Response 1: Thanks for your valuable attention and opinions.
Reviewer 2: The theoretical foundation on which the research was based was lacking and should be introduced in the Introduction and then significantly expanded in the literature review. The authors should take as a starting-point one or more sufficiently contrasted theories and apply them to this new context of analysis to justify the need to develop this new research. The paper should incorporate a more solid argumentation that allows justification of the reason for the selection of the explanatory variables that are considered in the empirical analysis. Stating that there is a lack of theoretical explanations and empirical evidence is an insufficient reason/justification to conduct the research. Furthermore, as far as possible, the theoretical framework should be sufficiently solid to justify that the relevant variables that should explain the phenomenon under study are those considered in the analysis and only those variables. In short, why is this research necessary; why are you using the theoretical model; what are the research gaps; and what recent justification have you provided for the aforementioned? In summary, fully explain the theoretical framework (in the Introduction and Literature review sections) that served as the foundation for your conceptual model that you developed.
-The contributions, introduction and literature review was suitably updated as requested in response to this comment.
Response 2: Thanks for your valuable attention and opinions.
Reviewer 3: Additionally, the Introduction section was much too short and did not cite a sufficient number of sources. The Introduction typically provides the background to the research problem and lays the groundwork by citing the origin of the idea/problem. There are generally several sections that provide a natural flow:
Overview / background information: Provide an overview / background information on all of the different variables of your research title in this section! In what context (industry area) does the problem find itself and the importance of the sector under investigation?
The Introduction section was aptly expanded and context was five to the overview / background information on all of the different variables were provided/highlighted.
Response 3: Thanks for your valuable contributions.
Reviewer 4: Area of concern / topic importance: Clarify the area of concern, or whatever needs to justify the research. Indicate why you believe that it is, in fact, a researchable problem, i.e. is the problem researchable and is it feasible? Do the preliminary data and available resources support its feasibility? Why is the topic important to discipline, country, region and globe?
-The area of interest/subject importance was expanded as recommended.
Response 4: Thanks for your comments.
Reviewer 5: Research gaps (as mentioned above): Have others studied this area - is there gap worth exploring? What are the research gaps, identified by others (in-text reference all sources, since this provides justification for your research? Use multiple in-text referencing/citations to collectively highlight a number of studies that identified potential gaps in terms of the topic.
-The research gaps were also expanded as suggested.
Response 5: Thanks for your valuable contributions.
Reviewer 6: Mandate more research is needed: Find journal articles/theses that mandate more research is needed in your area of study/topic by reading the “Introduction” and “Limitations and future research” sections, which will show possible research gaps / new research areas. Example of possible research gaps (all sources should be in-text referenced): Additional research should be conducted in different countries; different samples / target groups / unit of analysis; different industries / institutions; different research approaches (qualitative vs. quantitative); larger samples sizes / or more representative samples; new theoretical model, etc.! This section should be expanded upon throughout the literature review.
-New references were included and the research gaps were highlighted as well as potential risks in the literature according to my comment.
Response 6: Thanks for valuable contributions.
Reviewer 7: Lastly, end the “Introduction/background to the research problem” section with a statement that provides the main aim of your study that would serve to address the abovementioned research gaps.
-The research problem was appropriately updated as per my suggestions.
Response 7: Thanks for your valuable contributions.
Reviewer 8: Theoretical Framework (Literature Review) section was also lacking. The theoretical framework helps the researcher to explain time tested theories that embody the findings of numerous investigations on how phenomena occur under given scenarios. The theoretical framework can be written in prose (written format) and in a visual format (diagram) reinforced by a summarized explanation of the diagram. The theoretical framework frames the study and identifies the key theories and concepts that underpin your study. Stated differently, the theoretical framework provides background information on your study, which must emanate from a theoretical model in the field of study. Your discourse should explain the “how” and “why” things have transpired in the manner that they have, to anchor the research and thesis. Your narrative must include examples of the theoretical underpinning, where it has been used and by who, as well as a summary of their findings. Hence, your theoretical framework needs to be made more apparent, and your review of literature section should also be significantly expanded.
-The theoretical framework literature was aptly updated and expanded based on my suggestions.
Response 8: Thanks for your valuable contributions.
Reviewer 9: The was NO reference list? However, ensure that you provide a number of recent academic sources (e.g. journal articles) from 2020 – 2022, which should be incorporated throughout the paper to support the research, since there is much recent research on YouTube.
- The reference list was included as requested and the sources were sufficiently current.
Response 9: Thanks for your valuable contributions.
Reviewer 10: The Material and Methods section was apt.
-No response or action was required from the authors for this comment.
Response 10: Thanks for your valuable contributions.
Reviewer 11: The Results and Discussion sections were appropriately presented. However, I could not check the age of your sources since you omitted the reference list, so please be sure to supplement the Discussion section with recent literature sources as mentioned above if this is not the case.
-The age of my sources was sufficiently recent in this section.
Response 11: Thanks for your valuable contributions.
Reviewer 12: There Conclusions section was very brief, so you should also include a separate Theoretical Implication section and a separate Managerial or Practical Implications’ section as subheadings in the Conclusion section as follows:
- Theoretical implications: What contribution has the research has made to theory based on the theoretical foundation/model that you selected (or in your case did not select).
- Managerial or practical implications: Explain the contribution that the research has made in terms of managerial or practical implications.
-The Theoretical implications and Practical implications sections were expanded as suggested.
Response 12: Thanks for your valuable contributions.
Reviewer 13: I detected one spelling error, so it is suggested that you read through the paper again, as well as run it through a “Spelling and Grammar” check to establish if there other errors.
Page 13 line 461: “5.1. Theoritical Implications” should be “5.1. Theoretical Implications”
Response 13: Thanks for your valuable contributions. According to your suggestions, all of the article was updated in terms of “Spelling and Grammar”.
Reviewer 14: Overall, an interesting paper, which makes an original contribution, and I am generally satisfied with the author changes, so the paper can now be accepted for publication. However, it is important the one last a “Spelling and Grammar” check is conducted before the paper is published.
Response 14: Thanks for your valuable contributions. According to your comments, required corrections was performed.

Reviewer 2 Report
Please check this:
- while categorical variables were displayed as percentages (%). While categorical variables were displayed as percentages (%).
Author Response
Thanks for your valuable contributions, required corrections were performed.
